

# The effect of light on N₂ fixation and net nitrogen release of field

# *Trichodesmium*

Yangyang Lu[1,2], Zuozhu Wen[1,3], Dalin Shi[1,3], Mingming Chen[1,2], Yao Zhang[1,2], Sophie Bonnet[4], Yuhang Li[5], Jiwei Tian[6] and Shuh-Ji Kao[1,2*]

[1]State Key Laboratory of Marine Environmental Science, Xiamen University, Xiamen, China

[2]College of Ocean and Earth Sciences, Xiamen University, Xiamen, China

[3]College of Environment and Ecology, Xiamen University, Xiamen, China

[4]IRD, Aix Marseille Université, CNRS/INSU, Université de Toulon, Mediterranean Institute of Oceanography (MIO), New Caledonia, France

[5]Institute of Oceanology, Chinese Academy of Sciences, Qingdao, China

[6]Physical Oceanography Laboratory, Ocean University of China, Qingdao, China, Qingdao China

[*]*Correspondence to:* sjkao@xmu.edu.cn

**Abstract.** Dinitrogen fixation (NF) by marine cyanobacteria is a crucial pathway to replenish the oceanic bioavailable nitrogen inventory. Light is the key to modulate NF, however, field studies regarding light response curve (NF-I curve) of NF rate and the effect of light on diazotroph derived nitrogen (DDN) net release are missing that may hamper an accurate nitrogen model prediction. Uncontaminated $^{15}N_2$ gas dissolution method was applied to examine how the light change may influence the NF

intensity and DDN net release in the oligotrophic ocean. Experiments were conducted at stations with diazotrophs dominated by filamentous cyanobacterium *Trichodesmium spp.* in the Western Pacific and the South China Sea. The light effect on carbon fixation (CF) was measured in parallel using the $^{13}C$ labelling method specifically for a station characterized by *Trichodesmium* bloom. Both NF-I and CF-I curves showed $I_k$ (light saturation coefficient) range of 328 to 509 µE m$^{-2}$ s$^{-1}$ with saturation light at around 600 µE m$^{-2}$ s$^{-1}$. The proportion of DDN net release ranged from ~6% to ~50% revealing an increasing trend as the

light intensity decreased. At the *Trichodesmium* bloom station, we found CF/NF ratio was light-dependent and the ratio started to increase as light was lower than the carbon compensation point of 300 µE m$^{-2}$ s$^{-1}$. NF pathway was likely preferentially blocked under low light to conserve energy for photosynthesis, thus, there is a metabolism tradeoff between carbon and nitrogen fixation pathways under light stress. Results showed that short-term light change modulates the physiological state, which subsequently determined the C/N metabolism and DDN net release of field *Trichodesmium*. Energy reallocation

associated with the variations of field light intensity would be helpful for model prediction of global biogeochemical cycle involved with *Trichodesmium*.

**Keywords: diazotroph derived nitrogen release, Nitrogen fixation irradiance curve, *Trichodesmium***



## 1. Introduction

The bioavailable nitrogen introduced via NF by cyanobacteria is crucial to fertilize the tropical and subtropical oligotrophic surface ocean (Karl et al., 1997). In such environments, nitrate supply from the subsurface is generally limited by thermostructure induced stratification and NF can directly input bioavailable nitrogen to euphotic zone (Capone et al., 2005).

Among the variety of diazotrophs, the filamentous non-heterocystous cyanobacterium *Trichodesmium* is recognized as a major player, contributing to up to 80-110 Tg N annually, i.e. ~50% of global marine NF (Capone 1997). It often forms colonies or aggregates and under appropriate circumstances, forms large surface blooms (Zehr 2011).

Light is the ultimate energy source for the photoautotrophic diazotrophs and the energy-exhausting NF process is tightly linked with photosynthesis (LaRoche and Breitbarth, 2005 and reference therein). Regarding the light response of

10 *Trichodesmium*, several previous field studies put efforts on CF and oxygen production in response to irradiance (P-I curve) and showed that photosynthetic rates of *Trichodesmium* were proportional to light intensities, and that *Trichodesmium* have a relatively high irradiance requirement and a high respiration rate to protect the nitrogenase enzyme from $O_2$ deactivation (Lewis et al., 1988; Carpenter 1995;). By using the $C_2H_2$ reduction method, Carpenter et al., (1993) investigated the light response of nitrogenase activity for the field-towed *Trichodesmium*, which showed a response pattern as a function of irradiance and

15 resembling the P-I curve. Similarly, by using $^{13}C/^{15}N$ isotope labelling techniques, Holl et al., (2007) found that NF and CF rates of field-towed *Trichodesmium* were attenuated as light intensity decreased. In controlled laboratory experiments, Breitbarth et al., (2008) suggested that both nitrogenase activity and growth rates of *Trichodesmium* (IMS-101) are light-dependent (15 to 1100 $\mu E\ m^{-2}\ s^{-1}$), and Bell and Fu (2005) observed an increasing NF rates with the increase of light intensity (PAR 10–160 $\mu E\ m^{-2}\ s^{-1}$) and the cellular concentrations of Chl *a* and phycobiliproteins (PBPs) increased under low light

conditions.

Meanwhile, statistical analysis performed on the global dataset of field NF suggests that light is an important environmental factor explaining most the spatial variance of NF at the global scale (Luo et al., 2014). However, it has to be noted that some of the NF rate measurements available in this global database might be questionable due to previously unrealized technical problems, e.g., incomplete $^{15}N_2$ dissolution in the $^{15}N_2$ bubble labelling method (Mohr et al., 2010), bioavailable $^{15}N$ forms

contamination in some commercial $^{15}N_2$ gas (Dabundo et al., 2014) and inconsideration of diazotroph-derived N (DDN) release



in the filtrate fraction (Konno et al., 2010). Nevertheless, above mentioned experiments and global analysis support the idea of a light control on NF activity, CF and oxygen evolution of *Trichodesmium*; however, limited field experiments have been conducted on studying the light effect on C and N fixation of bulk seawater, particularly during naturally-occurring *Trichodesmium* blooms. Moreover, to our knowledge, no study has been implemented by using the improved $^{15}N_2$ dissolution

method (Mohr et al., 2010).

During the NF process, *Trichodesmium* release 10% to 50% of the $DD^{15}N$ in the dissolved pool (Glibert and Bronk, 1994; Konno et al., 2010), primarily as dissolved organic N (DON, such as dissolved free amino acid DFAA) and $NH_4^+$ (Capone et al., 1994; Mulholland et al., 2004). High DON and $NH_4^+$ concentrations are often measured within *Trichodesmium* blooms (Karl et al., 1992; Lenes et al., 2001), being supportive of DDN release. Most NF rates are measured via the incorporation of

$^{15}N$ into particulate organic N (PON) following incubation in the presence of $^{15}N_2$. The $^{15}N$ enrichment of the dissolved pool is generally not taken into account, resulting in aforementioned potential underestimation of NF rates. On the other hand, diatom and dinoflagellate blooms have been observed following *Trichodesmium* blooms, suggesting that DDN potentially supported non-diazotrophic phytoplankton growth (Devassy et al., 1978; Lenes et al., 2001). By using nanometer scale secondary ion mass spectrometry, Bonnet et al., (2016a) recently showed that the DDN is quickly (1-3 days) transferred to surrounding

plankton, predominantly diatoms and bacteria, during *Trichodesmium* blooms. A mesocosm experiment performed in the Western Tropical South Pacific (VAHINE) revealed an incommensurately high contribution of NF to export production (>50 %, Knapp et al., 2016) during a bloom of UCYN-C bloom. This export was largely indirect, i.e. attributable to quick recycling processes of DDN transfer to non-diazotrophs that were subsequently exported (Bonnet et al., 2016b; Bonnet et al. 2016c; Knapp et al., 2016). In spite of the importance of DDN release in C and N cycles, the factors controlling *Trichodesmium* DDN

release remained unclear. In particular, the effect of light on DDN release has been poorly studied. To date, only one study reports a significant release of DDN and DOC in culture after a rapid shift from low-light to high-light regimes to protect the photosynthetic apparatus (Wannicke et al., 2009).

Here we investigated the effect of light on DDN release and C/N fixation stoichiometry of *Trichodesmium* in the field under contrasting situations, i.e. during a *Trichodesmium* bloom in the Western Equatorial Pacific and in a non-bloom area in

the South China Sea.

## 2. Material and Methods

This study was performed onboard the R/V Dongfanghong II during two cruises to the Western Equatorial Pacific Ocean (06 December 2015 to 12 January 2016) and the South China Sea (15 May to 07 June 2016). Experiments were conducted at three stations (Supplementary information Fig.1), among which one of them was characterized by the presence of a

*Trichodesmium* bloom (Western Equatorial Pacific Ocean Sta. S0320), the other two were located at South China Sea (A3, D5).

### 2.1. Seawater sampling and experimental procedures

Water samples were collected from 3-5 m depth using 10 L Go-Flo bottles which were attached to a CTD rosette (Seabird 911 CTD). In our experiments, same 4.5L surface water samples were collected in the polycarbonate (PC) bottles and then put

in six on deck incubators with different light intensities for NF rate incubations. The light source was natural solar irradiance and light intensity gradients (92%, 54%, 28%, 14%, 8%, 1% of surface irradiance) were manipulated by using neutral density screen to adjust the light level. During the incubation period, the light intensity was monitored on-deck with a $2\pi$ photosynthetically available radiation (PAR) sensor (LI-1400; LI-COR). We took the average light intensity of incubation light period as the surface irradiance to calculate light intensities of the six light gradients.

### 2.2. Nutrients, Chl *a* and *Trichodesmium* abundance

Nutrient samples were collected in 100ml high density polyethylene (HDPE) bottles and kept frozen at -20 °C freezer until analysis. Nanomolar levels of SRP were determined according to Ma et al., (2008) with a detection limit of 1.4 nM and relative precision of $\pm$ 2.5%. Nanomolar levels of nitrate were analyzed by chemiluminescent method (Garside, 1982) with a detection limit of 2 nM.

For Chl *a* concentrations determination, 1 L of seawater was filtered on GF/F filters, wrapped in aluminum foil and stored at –20°C until analysis onshore. Chl a was extracted in 90% acetone refreezing for 24 h and analysed fluorometrically according to method described by Welschmeyer 1994.

For *Trichodesmium* abundance determination, 1 L of seawater was sampled in HDPE bottles and immediately fixed with 10 mL Lugol's solution. Onshore, subsamples were settled for 48 h, the supernatant was removed and *Trichodesmium* filaments

(trichomes) were counted on a Nikon Eclipse 50i optical microscope.

### 2.3. Molecular assessment of diazotrophs

For DNA analysis, 4 L of seawater were filtered through 0.2 μm pore-sized membrane filters (Supor-200, Pall Gelman, NY, USA) which were stored in a liquid nitrogen until analysis. DNA was extracted according to (Massana et al., 1997) with some modifications. Briefly, each filter was cut into pieces and placed into a 2ml sterile screw cap micro tube containing 0.2 g autoclaved glass beads and 0.8ml GTE buffer (100 mM EDTA, 50 mM Tris, 0.75M sucrose). The tubes were agitated three

5    times for 40s in a homogenizer (FastPrep-24, MP Bio, USA) at 4.5m/s, then froze-thaw three times in liquid nitrogen. The next steps followed the protocol of (Massana et al., 1997).

Four published quantitative Polymerase Chain Reaction (qPCR) probe–primer sets (Church et al., 2015a, 2015b) were used for qPCRanalysis. Relevantly, the *nif*H genes of four photoautotrophic diazotroph groups were targeted: *Trichodesmium* spp., *Richelia* spp. associated with *Rhizosolenia* spp. (het-1), and the unicellular groups A (UCYN-A) and B (UCYN-B). We

used the thermal cycling conditions and reaction mixtures as described previously by Zhang et al., (2011) with slight modifications. Triplicate 20ul-QPCR mixtures were used for each sample and standard, reaction mixes contained 10ul Premix Ex Taq (Probe qPCR) (RR390A, Takara Bio Inc, Dalian, China), 400 nM each of forward and reverse primer, 400 nM of fluorogenic probe, and 1 μL of environmental DNA or plasmid standards. We used dilution series of four linearized plasmids as standards, which contained inserts matching four primer-probe sets respectively. The Real-time Quantitative PCR was

performed on an CFX96 Real-Time System (Bio-Rad Laboratories, USA) with the following thermal cycling conditions: 50°C for 2 min, 95°C for 2 min, and 45 cycles of 95°C for 15 s, followed by 60°C for 1 min. The quantification limit was determined empirically to be 1 copy per reaction. The amplification efficiency varied between 90% and 100%. The negative controls contained complete reaction ingredients except environmental DNA or standards, no amplification was found in negative controls.

**2.4. N$_2$ and carbon fixation rate measurements**

NF rates were determined according to the dissolution method: the $^{15}$N$_2$ enriched seawater was prepared following the same device and procedure as described in Shiozaki et al., (2015) and 200 mL $^{15}$N$_2$-enriched seawater was added into each 4.5L PC incubation bottle. The $^{15}$N$_2$ gas (98.9%) by Cambridge Isotope Laboratories was used. We conducted blank check for $^{15}$N$_2$ gas (contamination of bioavailable non-N$_2$ $^{15}$N) as mentioned in Dabundo et al., 2014. Briefly, triplicate 2 mL $^{15}$N$_2$ gas

and 10 mL natural seawater were injected to 20 mL headspace vials, sealed with septum stopper, and then shaken overnight.



The $\delta^{15}N$ of TDN was measured and compared with the $\delta^{15}N$ of natural seawater samples. Values of $\delta^{15}N$ TDN of blank seawater and test seawater group were 4.7‰ and 5.0‰, respectively, revealing no contamination of the $^{15}N_2$ gas.

At Sta. S0320, the *Trichodesmium* bloom station, $^{13}C$-labeled sodium bicarbonate (99 atom% $^{13}C$; Cambridge Isotope Laboratories) was added in parallel with $^{15}N_2$ to each bottle at a final tracer concentration of 70 μmol L$^{-1}$ to simultaneously measure the CF and NF rates. At each irradiance level, triplicate water samples (4.5L PC bottle) were incubated on-deck in incubators with surface seawater flow through.

After 24h incubation, water samples were gently filtered (<200mm Hg) onto pre-combusted (450℃, 4 h) 25 mm Whatman GF/F filters, preserved at -20℃ and then dried at oven over night (50℃). The POC/PON concentrations and isotopic values were analysed on a Flash EA (Thermo Fisher Flash HT 2000)-IRMS (Thermo Fisher Delta V plus). International reference material (USGS40) with certified $\delta^{15}N$ and $\delta^{13}C$ value of -4.5‰ and -26.2‰, respectively, was inserted every 8 samples to check the drift and ensure the accuracy of the measurements. The reproducibility for $\delta^{15}N$ and $\delta^{13}C$ measurements were both better than 0.3‰. The NF and CF rates were calculated by using similar equations proposed by Montoya et al., (1996) and Hama et al., (1983), respectively.

### 2.5. Light-response curves for N$_2$ fixation and carbon fixation

Follow the photosynthetic model by Webb et al., (1974):

$$N = N_m(1 - \exp(-\alpha I / N_m)) + N_d , \qquad (1)$$

Where $N_m$ is the maximum rate of NF at light saturating irradiance, $N_d$ is the rate measured in darkness, $I$ is the natural irradiance and α is the light affinity coefficient for NF rate, $I_k$ stands for the light saturation coefficient ($N_m /\alpha$), we constructed the irradiance curve for NF. Similarly, the light response curve of CF was obtained.

### 2.6. DDN net release to the dissolved pool

40 mL of the filtrate (passed through pre-combusted GF/F filters) of each NF incubation bottle was collected and preserved at -20℃ to determine the TDN concentration and $\delta^{15}N$-TDN according to Knapp et al., (2005). Briefly, TDN was oxidized to nitrate by persulphate oxidation reagent (purified by recrystallization 3-4 times) and the concentration was measured by the chemiluminescent method (Garside, 1982). The $\delta^{15}N$-TDN-derived nitrate was analyzed by using the 'denitrifier method' (Sigman et al., 2001). The reproducibility for $\delta^{15}N$-TDN measurements was better than 0.5‰. The DDN released to the





dissolved pool was calculated following the equation proposed by Bonnet et al., (2016a).

**2.7. Transfer of DDN into non-diazotrophic plankton**

To evaluate the short time (24h) DDN transfer to non-diazotrophic plankton, we followed the method by Adam et al., (2016). Briefly, for the control group, 10μm sieve was used to remove most *Trichodesmium* colonies and the remaining

community was incubated for 24h with $^{15}N_2$-enriched seawater. In another group, the whole community was incubated for 24 h and *Trichodesmium* colonies were removed after incubation terminated. Each experiment was performed in triplicates. The $\delta^{15}N$ difference between the two treatments was considered to be a proxy of the DDN transfer to non-diazotrophic plankton.

**3. Results**

**3.1. Environmental conditions**

The temporal patterns of PAR were shown in Figure 1. The sun rose at ~6 AM and set at ~6 PM. Value of PAR (sampling at ten second interval) varied rapidly in a wide range from 0 to 5000 μE m$^{-2}$ s$^{-1}$, which are the classical values observed at low latitudes, yet much higher than those generally used in laboratory culture experiments (Bell and Fu 2005; Wannicke et al., 2009). Although incubations were conducted for 24 h, average PAR during the incubation period (light intensity > 1μE m$^{-2}$ s$^{-1}$) were applied for discussion. The average PAR values were 2185, 2212 and 1200 μE m$^{-2}$ s$^{-1}$ for Stations S0320, A3 and D5,

respectively.

The hydrographic and biogeochemical parameters are shown in Table 1. All three stations were characterized by low nutrient concentrations (NO$_3^-$ 6 to 11 nM, PO$_4^{3-}$ 13 to 100 nM), high salinity (34.5-34.6) and high sea surface temperature (27.6-29.7℃). At the *Trichodesmium* bloom station (Sta. S0320), Chl *a* concentrations were 1.2 mg m$^{-3}$, much higher than

those measured at the other stations (0.25, 0.39 mg m$^{-3}$, respectively). Result of the *nif*H phylotype abundances showed that *Trichodesmium* accounted for >98.8%, 88.6% and 96.4% of the diazotrophic community in Sta. S0320, A3 and D5, respectively (Fig.2), thus *Trichodesmium* was the dominant cyanobacteria diazotroph in all experiments. The dominant *Trichodesmium* species were *Trichodesmium thiebautii* for Stas. S0320 and D5, with abundance of 4227 ± 679 (n=6), and 190 ± 50 (n=6) trichomes L$^{-1}$, respectively. POC /N concentration of the <10μm fraction represented <25% of the bulk POC/N (see

Table 2 and below), supporting that *Trichodesmium* was the dominant phytoplankton community at the blooming station.



### 3.2. Light response of net (particulate) N₂ fixation

The net NF rates at the surface light intensity were $390.6 \pm 20.4$, $12.2 \pm 1.8$, $9.9 \pm 0.4$ nM N d$^{-1}$ at Sta. S0320, A3 and D5,

respectively. The NF rate at the blooming station was two orders of magnitude higher than that of at the two non-bloom

stations. Detail experimental data, including concentrations and isotopic values, for initial and final time points were listed

in supplementary information Table 1-3. However, trichomes-normalized rates were 92 and 52 pM N trichomes$^{-1}$ d$^{-1}$,

respectively, for Stas. S0320 and D5 revealing a more consistent rate per biomass.

As shown in Figure 3, these NF-I curves showed a general pattern indicating that net NF rates increased significantly with

light intensity from 10 to 600 µE m$^{-2}$ s$^{-1}$, and then saturated at around 600 µE m$^{-2}$ s$^{-1}$. The simulated $I_k$ values for NF were

404, 328 and 509 µE m$^{-2}$ s$^{-1}$, respectively, for Stas. S0320, A3 and D5 with an average value $414 \pm 90$ µE m$^{-2}$ s$^{-1}$.

### 3.3. Particulate C/N metabolism of *Trichodesmium* bloom

Results of CF showed a traditional P-I curve pattern without apparent light inhibition (solid curve in Fig. 4a). The fitted

curve of CF showed consistent pattern with those of NF (dashed curve in Fig. 4a) giving an $I_k$ value of 455 µE m$^{-2}$ s$^{-1}$ falling

within the $I_k$ range for the three NF-I curves. However, the ratio of CF to NF was variable as light varied (Fig. 4b). The ratio

ranged from $7.4 \pm 0.6$ to $9.3 \pm 1.0$ when light intensities were saturating while the ratio increased significantly from $7.4 \pm 0.6$

to $16.8 \pm 3.2$ as light intensities decreased from 612 to 22 µE m$^{-2}$ s$^{-1}$.

The initial concentrations (n=3) of POC and PON were $13.4 \pm 0.1$ µM, $2.1 \pm 0.0$ µM, respectively, with a mean C:N molar

ratio of 6.4 (horizontal lines in Fig. 4c an d), which is almost identical to the Redfield C/N ratio of 6.6. After incubations under

various light intensities, the final POC concentrations showed a decreasing trend ranging from $17.3 \pm 1.2$ µM to $10.9 \pm 1.0$ µM

as the irradiance decreased. Below ~300 µE m$^{-2}$ s$^{-1}$, the final POC concentration was even lower than the initial POC

concentration (red dashed line in Fig. 4c) suggesting that the light compensation point ($I_c$) is around 300 µE m$^{-2}$ s$^{-1}$. Similar

light dependent pattern was found for PON, yet, final PON concentrations, varying from $2.1 \pm 0.2$ µM to $2.5 \pm 0.2$ µM, were

always higher than the initial concentration (blue dashed line in Fig. 4c) without compensation point.

The observed C:N ratio of bulk particulate matter (5.3-7.0;Fig. 4d) is consistent with previously reported ranges for

*Trichodesmium* (LaRoche and Breitbarth, 2005; Mulholland, 2007). However, a strong light dependency was observed also

for the final C/N after incubation. The saturated irradiance of ~ 600 µE m$^{-2}$ s$^{-1}$ was likely a threshold, below the saturation light



the final C/N tended to be lower than initial C/N of 6.4 (dashed horizontal line in Fig. 4d).

### 3.4. DDN net release to the dissolved pool

The rate of $DD^{15}N$ net release in the TDN pool ranged from $7.7 \pm 0.4$ to $54.1 \pm 7.8$ nM N $d^{-1}$ for Sta. S0320, from $0.7 \pm$

0.2 to $1.0 \pm 0.1$ nM N $d^{-1}$ for D5, and from $1.9 \pm 1.5$ to $5.0 \pm 1.6$ nM N $d^{-1}$ for A3. The contribution of DDN net release to gross

5 NF ranged from $8\% \pm 0\%$ to $25\% \pm 6\%$, $6\% \pm 6\%$ to $45\% \pm 14\%$ and $14\% \pm 11\%$ to $50\% \pm 5\%$ for Stas. S0320 D5 and A3,

respectively (Fig. 5), which agrees well with previous field studies (Glibert and Bronk, 1994; Mulholland et al., 2006; Bonnet

et al., 2016a; Konno et al., 2010; Benavides et al., 2013; Berthelot et al., 2015). Our data revealed that the fraction of DDN

release to gross NF was increasing as light decreased.

### 3.5. DDN transfer to non-diazotroph biomass

10 After 24 h incubation, the DDN transfer rates (transferred to the non-diazotrophic plankton) were $18.6 \pm 3.6$, $0.5 \pm 0.3$ and

$0.7 \pm 0.5$ nM N $d^{-1}$ corresponding to $5\% \pm 1\%$, $4\% \pm 3\%$ and $5\% \pm 4\%$ of total NF (net plus dissolved), respectively, for Stas.

S0320, D5 and A3 (Fig. 6). Our fractions are consistent with previous reports by Bonnet et al., (2016a), in which $6\% \pm 1\%$ of

$DD^{15}N$ was transferred to non-diazotrophic plankton in naturally occurring *Trichodesmium* blooms and slightly lower than

$DD^{15}N$ transfer (~12%) by Berthelot et al., (2016) who inoculated *Trichodesmium. erythraeum* into natural surface oligotrophic

15 seawater. Our results confirm that *Trichodesmium* could actively transfer newly fixed nitrogen to non-diazotrophs.

## 4. Discussions

### 4.1. High light demand for *Trichodesmium* N₂ fixation.

The simulated $I_k$ values for NF locates at the high end of the reported $I_k$ values for photosynthesis (LaRoche and Breitbarth,

20 2005). These values suggest a high light demand for *Trichodesmium* NF. The high energy requirement of *Trichodesmium* is

not only for breaking the strong of triple bond of the N₂ molecule, but also for numerous strategies, such as high respiration

rates and the Mehler reaction, to protect the sensitive nitrogenase against the oxygen evolved by photosynthesis during day

time (Kana 1993). Thus, *Trichodesmium* is generally dwelled in the upper euphotic zone of tropical and subtropical ocean to

meet the high light demands (Capone et al., 1997).

25 Generally, in the tropical and subtropical regions, average surface light intensities are around 2000 $\mu E$ $m^{-2}$ $s^{-1}$ in sunny days.

By taking into account light extinction coefficient of seawater, we calculated that the optimal depth for *Trichodesmium* to hold

maximum NF rate would be shallower than 15-40m . This result matches well with many field observations that most NF had

occurred in the well-lit (0-45m) region of the euphotic zone (Capone et al., 1997; Böttjer et al., 2016). This also agrees well

with the observation that maximum *Trichodesmium* densities often appears at around 15 m depth and typically forms bloom

in surface (Carpenter and Price 1977; Capone et al., 1997).

Our results also suggest that NF of *Trichodesmium* could respond to variable light intensity in the field within a short time

period (24h). Such result means that light conditions during on-deck incubations should also be presented along with rate data

if we want to compare field NF results among different studies. Unfortunately, the field NF rates had rarely been reported with

consideration of *in situ* light conditions although the light control on NF is well known to researchers.

Compared with laboratory strains acclimated to low light, field observed NF-I curves are more representative of real

ocean with greater applicability. The parameter consistency among our three stations in NF-I curves regardless the wide range

of trichomes biomass and maximum NF rates, offers critical information for light-associated parameters in model predictions

of global nitrogen fixation (Fennel et al., 2001; Hood et al., 2001).

**4.2. Metabolism tradeoff between carbon and nitrogen fixation under light stress**

In our field incubations, bulk C/N molar ratios were always lower than the corresponding net CF:NF ratios at all light

intensities (Fig.4b, 4d). As reported in both culture and field studies, *Trichodesmium* usually exhibits a higher CF:NF ratio

than expected stoichiometric value of 6.6 (Mulholland, 2007). Several hypotheses have been proposed: 1) the underestimation

of gross NF rates by overlooking the [15]N signal in dissolved pool (Glibert and Bronk, 1994; Mulholland et al., 2004), 2) the

underestimation of N assimilation rate if there is uptake of other N sources such as nitrate or ammonium (Mulholland et

al.,1999), 3) high carbon requirements to synthesize carbohydrate as ballast for vertical migration (Villareal and Carpenter,

1990;), 4) the support of the high energy-cost high respiration and Mehler reaction pathways (Carpenter and Roenneberg,

1995), 5) the CF by non-diazotrophic phytoplankton.

Here, the low DDN net release rate is not supportive of the first hypothesis. As the incubation experiments were used the

same bulk water and only light intensity was manipulated, the initial bioavailable nitrogen concentration between different

treatments almost the same, so no apparent evidence support second hypotheses. Meanwhile, the third and fourth could not



explain the increased CF:NF ratio trend with the decrease of light intensity over the low light condition. In fact, the contribution from non-diazaotrophic phytoplankton to CF cannot be excluded during bulk water incubation, however, the contribution is limited even at low light after assessment (see Supplementary information). As aforementioned, *Trichodesmium* was the dominant phytoplankton species, thus, the variation pattern of CF rates and POC concentrations against different light intensity

mainly reflects the carbon metabolism of *Trichodesmium*.

Under light limitation, *Trichodesmium* faced sever carbon consumption and energy shortage, energy was likely reallocated between CF and NF. We hypothesized that under low light stress, *Trichodesmium* physiologically prefer to sacrifice NF to save more energy for CF to alleviate the intensive carbon consumption by respiration. This is analogous to the *Trichodesmium* iron limitation metabolism, of which photosynthesis take the priority over NF to get iron (Shi et al., 2007). Since our incubations

were conducted in short term, such metabolism tradeoff between carbon and nitrogen fixation may happen frequently and widespread in the field for *Trichodesmium* under low light conditions, such as cloudy day, rainy day and even in deeper water with less light.

### 4.3. Light modulation of DDN net release fraction

In fact, previous study found that *Trichodesmium* trichomes contain only 15–20% of diazocytes cells capable of NF (Kranz

et al., 2011 and reference therein). The remaining non-diazocytes cells rely on the release of bioavailable N, mainly the form of ammonium or amino acid, from diazaocytes (Mulholland et al., 2004; Kranz et al., 2011). This process is directly proved by $^{15}N$ labelling and Nano-SIMS method in which the $^{15}N$ signal is rapidly distributed into the majority cells of *Trichodesmium* trichomes and even the $^{15}N$ label signal is relatively lower in the center cells which probably a zone of diazocytes (Finzi-Hart et al., 2009; Bergman et al., 2013).

In this study, the increased proportion of DDN in the dissolved pool as the decrease of light intensity suggested that physiology status of diazotrophs takes control on the DDN release process. At station A0320 high light intensities ($> 600\ \mu E$ $m^{-2}\ s^{-1}$), the final POC and PON concentrations increased significantly also implying an active physiology status of *Trichodesmium* and the fraction of $DD^{15}N$ release in the dissolved pool ranged from 6% ± 6% to 23% ± 5%. Actually, several unialgal cultures studies, including *Trichodesmium* and UCYN-B and UCYN-C, showed less than 2% $DD^{15}N$ release in the

dissolved pool (Berthelot et al., 2015; Benavides et al., 2013). These low values were attributable to the exponential growth

phase and optimal growth conditions and lack of exogenous factors influence such as viral lysis (Hewson et al., 2004) and

sloppy feeding (O'Neil et al., 1996). Nevertheless, our values at high light are congruent with the field study (7–17 %) by

Berthelot et al., (2106). Similar to their finding, we suggested the active cell status and exposure to the exogenous factor may

only lead to slightly higher proportion DDN net release. Under the light limitation stress, the inactive physiology state condition

of *Trichodesmium* was reflected by the decrease of POC concentrations and activity of the CF and NF, thus, the DDN fixed by

diazocytes was likely not efficiently transferred to other cells along the trichomes therefore accumulating in the dissolved pool.

Furthermore, a part of cells could breakdown and directly releases intracellular bioavailable nitrogen. The fraction of DD$^{15}$N

release in the dissolved pool ranged from 17% ± 4% to 50% ± 5% at low light conditions (600 μE m$^{-2}$ s$^{-1}$). This conclusion is

also consistent with Bonnet et al., (2016a) for two natural *Trichodesmium* bloom studies that in the decaying bloom case, high

ammonium concentration accumulation (3.4 μmolL$^{-1}$) and high proportion of DDN release (20 ± 5 to 48 ± 5%) was observed,

while in the exponentially growing bloom case, the proportion of DDN release only ranged from 13 ± 2 to 28 ± 6% and without

apparent accumulation of ammonium.

As summarized in Berthelot et al., 2015, most of the higher end of reported DDN net release values were estimated by the

difference between gross NF rates measured by acetylene reduction assays (ARA) and the net NF measured by the $^{15}$N$_2$ bubble

labelling technique (Montoya et al., 1996). The known uncertainty of conversion factor for acetylene to N$_2$ for ARA method

(Shiozaki et al., 2010) and underestimation of net NF by the $^{15}$N$_2$ bubble method may potentially led to higher DDN net release

and biased estimates. In this study the direct measurement of the DD$^{15}$N in dissolved pool by the improved dissolution $^{15}$N$_2$

enriched seawater method (Mohr et al., 2010) was applied to assess the DDN net release, so our data were quite reliable.

## 5.  Conclusions

Regarding the light response curves of *Trichodesmium*, most studies have concentrated on the photosynthesis behavior

by using the oxygen evolution or $^{14}$C/$^{13}$C assimilation measurement. In this study, we provide quantitative information on light

effect on NF and DDN net release of field *Trichodesmium* and reveal that the NF was a function of light intensity and biomass.

The light requirement of *Trichodesmium* NF was high relative to its photosynthesis light demand. The empirical $I_k$ value

suggests *Trichodesmium* population maxima should appear at <15 m depth to obtain sufficient light energy. Furthermore, diel



light cycle is a crucial parameter to drive physiological state of *Trichodesmium*, which subsequently determined the C/N metabolism and DDN net release. Accordingly, we suggest the necessity to provide field light data along with nitrogen fixation data obtained via on-deck incubation for the future studies.

Recently studies revealed that unicellular cyanobacteria diazotrophs, inhabit different niches, especially UCYN-A, distributed more widely in global ocean and may contribute equal NF flux with *Trichodesmium* (Zehr et al., 2016; Martínez-Pérez et al., 2016). More field studies are needed in future to explore the light response of those UCYN to better understand their light behavior and to complete the role of diazotroph in global NF models.



Table 1. Environment condition of three stations surface water. nd: not determined

| Station | Salinity | Temperature (℃) | chl $a$ ($\mu$g L$^{-1}$) | SRP (nM) | NO$_X$ (nM) | DON ($\mu$M) | *Trichodesmium* Colonies (trichomesL$^{-1}$) |
|---------|----------|-----------------|---------------------------|-----------|---------------|----------------|----------------------------------------------|
| S0320 | 34.5 | 29.7 | 1.2 | 100 | 6 | 9.8 (1.2) | 4227 ± 67 (n=6) *thiebautii* |
| A3 | 34.6 | 27.6 | 0.39 | 13 | 7 | 7.2 (0.5) | nd |
| D5 | 34.6 | 29.3 | 0.25 | 24 | 11 | 8.4 (0.3) | 190 ± 50 (n=6) *thiebautii* |



Table 2. Synthesis of PON, POC and DON concentration, C/N and corresponding NF and CF rate, NF/PP in station S0320. Where the '<10μm-a' represent NF rate of <10μm

community incubated with > 10μm *Trichodesmium* colonies, '<10μm-b' represent the background NF rate of <10 μm community.

| irradiance | PON | Particulate NF rate | Dissolved NF rate | POC | Particulate CF rate | NF/PP | C/N |
|---|---|---|---|---|---|---|---|
| ($\mu$E m$^{-2}$ s$^{-1}$) | ($\mu$M L$^{-1}$) | (nM L$^{-1}$ d$^{-1}$) | (nM L$^{-1}$ d$^{-1}$) | ($\mu$M L$^{-1}$) | ($\mu$M L$^{-1}$ d$^{-1}$) | | |
| Initial condition | 2.1 (0.0) | - | - | 13.4 (0.1) | - | - | 6.4 (0) |
| 2010 | 2.5 (0.19) | 391 (20) | 32 (1.3) | 17.3 (1.2) | 3.6 (0.19) | 9.3 (1.0) | 7.0 (0.1) |
| 2010 (<10μm-a) | 0.55 (0.05) | 25.1 (3.2) | - | 4.1 (0.2) | 0.28 (0.03) | 11.6 (1.0) | 7.5 (0.1) |
| 2010 (<10μm-b) | 0.5 (0.1) | 6.5 (1.6) | - | - | - | - | - |
| 1180 | 2.3 (0.07) | 430 (39) | 47 (6.2) | 15.7 (0.13) | 3.4 (0.1) | 7.8 (0.7) | 6.8 (0.2) |
| 612 | 2.5 (0.09) | 401 (40) | 54 (7.8) | 15.8 (0.87) | 2.9 (0.08) | 7.4 (0.6) | 6.3 (0.1) |
| 315 | 2.4 (0.25) | 235 (10) | 50 (15) | 13.9 (0.93) | 2.0 (0.07) | 8.6 (0.5) | 5.9 (0.3) |
| 192 | 2.3 (0.09) | 85 (23) | 13 (3.3) | 12.6 (0.28) | 0.98 (0.26) | 11.6 (0.5) | 5.6 (0.4) |
| 22 | 2.1 (0.22) | 27 (8) | 7.7 (0.35) | 10.9 (1.0) | 0.44 (0.08) | 16.8 (3.2) | 5.3 (0.1) |





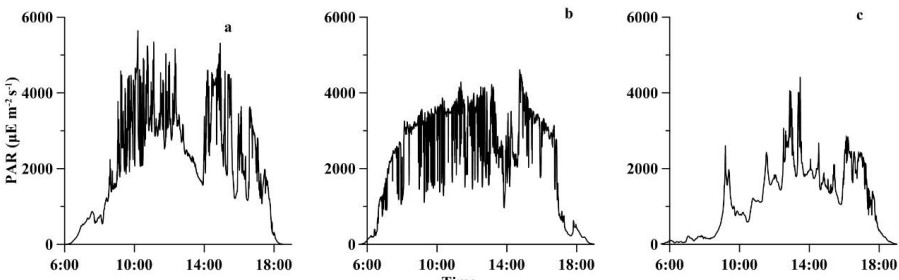

**Figure 1.** Temporal variations in photosynthetically active radiation (PAR $\mu E$ m$^{-2}$ s$^{-1}$) obtained on deck during the

experiment periods, a) for station S0320; b) for station A3; c) for station D5.




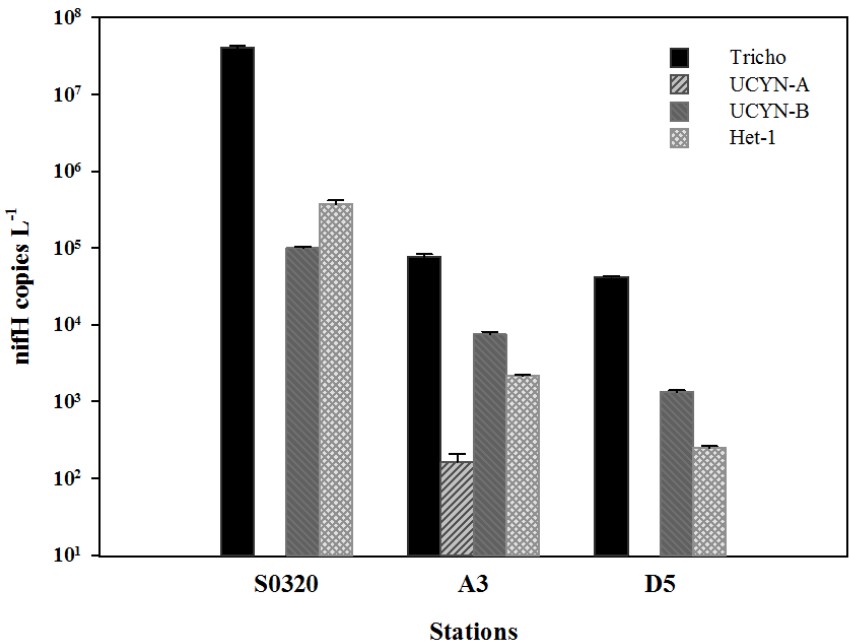

**Figure 2.** Cyanobacteria diazotrophs *nif*H phylotype abundances (*nif*H gene copies $L^{-1}$). 'Tricho' = *Trichodesmium*

spp.; 'UCYN' = unicellular $N_2$-fixing cyanobacteria from Group A, B; 'Het-1'= heterocystous cyanobacteria from

Group 1. Error bars represent the standard deviation for triplicate natural samples.





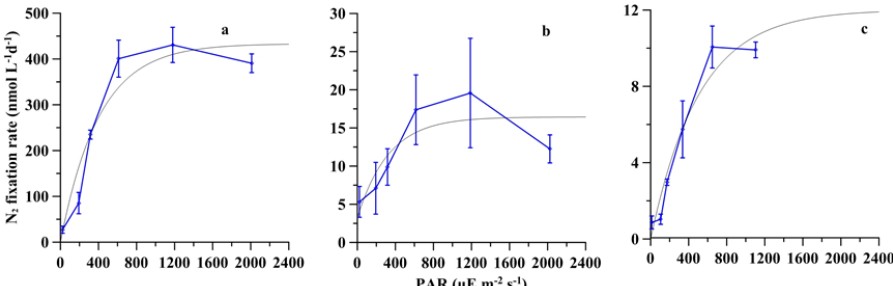

**Figure 3.** Net (particulate) NF versus irradiance. The gray curves represent the fitted NF-I curves. Error bar

represents the standard deviation of triplicate incubations. a) for station S0320; b) for station A3; c) for station D5.





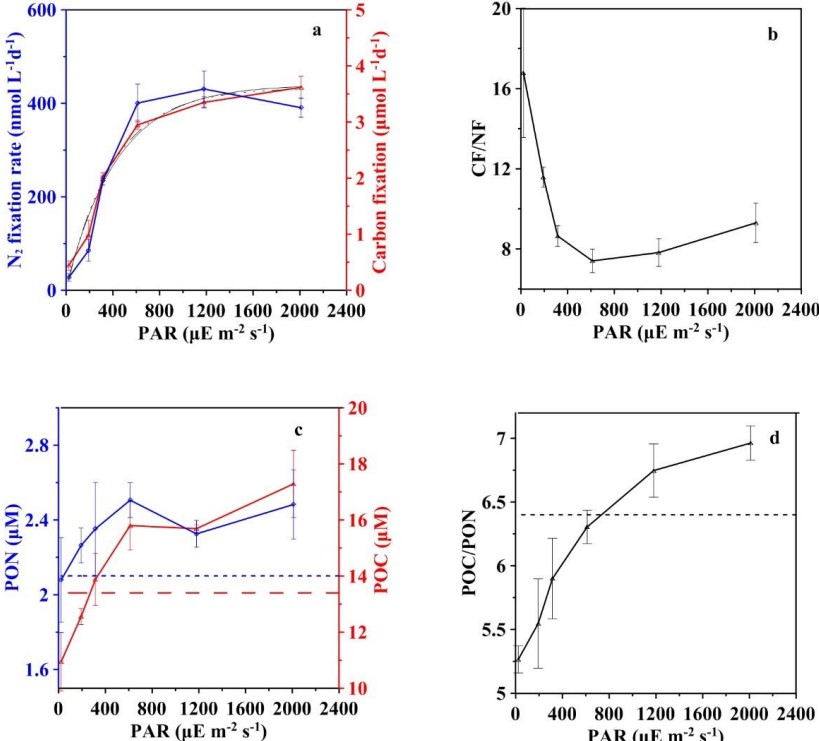

**Figure 4.** Light effect on carbon and nitrogen of community at Station S0320 with *Trichodesmium thiebautii*

bloom. (a) particulate Primary production (red solid line), NF (blue solid line) at different light intensity, fitted

light response curves for    PP (black solid line) and NF (black dotted line) (b) PP/NF ratio responded to different

light intensity; (c) The final concentration of POC (red solid line), PON (blue solid line) at different light intensity

after incubations, and initial POC (red dashed line), PON (blue dashed line) concentration; (d) C/N ratio at

different light intensity after incubations. Error bars represent the standard deviation on triplicate incubations.



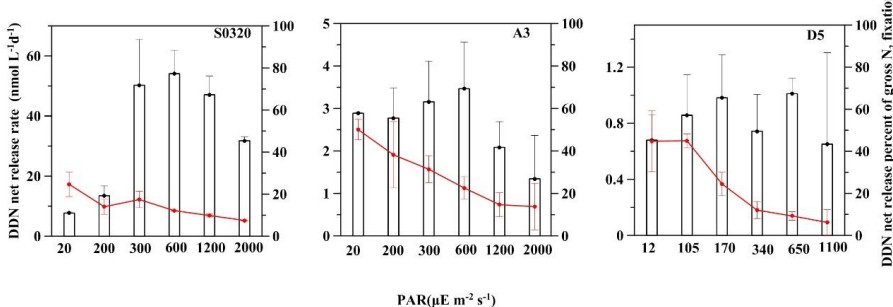

**Figure 5.** DDN net release rate (bar charters) and percentage of total NF (red lines) under different light intensities

at stations S0320, A3 and D5. Error bars represent the standard deviation on triplicate incubations.

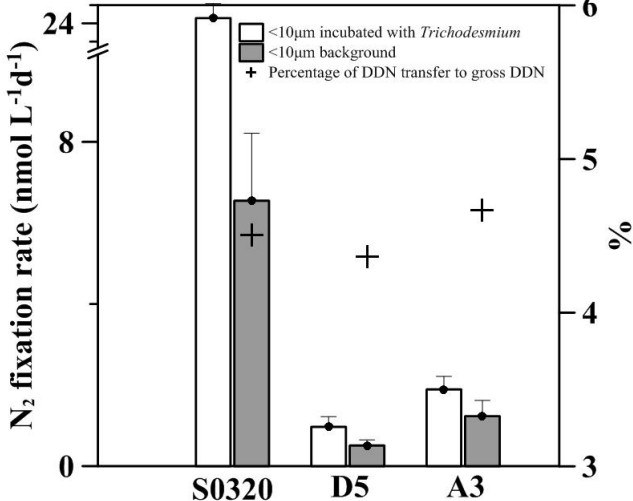

**Figure 6.** NF and DDN transfer measured in two treatment groups for Stations S0320, D5 and A3. The black bars

represent background NF rate of <10μm community. White bars represent NF rate of <10μm community incubated

with > 10μm *Trichodesmium* colonies. Error bar represent the standard deviation of triplicate.



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
