# Peer review of "Light effect on N2 fixation and net nitrogen release of *Trichodesmium"

_Biogeosciences, 2017_

## Short Comment (SC1) · 27 Jun 2017

From the perspective of a modeller with an interest in Trichodesmium, this is a useful paper. One reference you may have missed is Oliver et al. (2012). It would be interesting to see a comparison of the results in your manuscript with the results you'd expect given the energetic cost of nitrogen fixation versus nitrate uptake, and the energetic cost of chlorophyll production. Oliver et al. (2012) would be a useful resource to help make that comparison.

Oliver, R.L., Hamilton, D.P., Brookes, J.D. and Ganf, G.G., 2012. Physiology, blooms and prediction of planktonic cyanobacteria. In Ecology of cyanobacteria II (pp. 155-194). Springer Netherlands.

---

## Referee Comment (RC1) · Anonymous Referee #1 · 17 Jul 2017

Lu et al. have made isotopically enriched on-board incubation experiments to understand the role of light on N2 fixation. They further present estimates of nitrogen release from the diazotrophs and infer that light also plays a role on nitrogen release. Most of their findings are not new, have been known for some years now. The effect of light on nitrogen release seems interesting, but it is difficult to understand the way it's presented. I have provided several comments below that might improve the manuscript.

Major comments:

1. Diazotroph derived nitrogen (DDN) release increases with increase in light intensity at S0320 (Fig. 5 a) but there was no variation at the other two stations (Fig. 5b,c). However, % of the total NF release always decreases. On the other hand, N2 fixation increases with increase in light intensity and saturates at some point (Fig. 4a). Put all

these pieces of information together, it appears that it becomes difficult to say what role light play in DDN release. Diazotrophs would release N anyways, so what is the role light (they would release even if put them in absolute dark). Therefore, the discussion provided in the section 4.3 is not convincing.

2. Were all the incubations samples at each station taken from the same Niskin Bottle? I believe not because of bottle capacity. As we know the sample (especially when the cell abundance is high) in different Niskin bottles could not be homogeneous although taken from the same depth and the same CTD. If the cell abundances were different in different light incubations to start with, then the rates would be different because of cells and not because of light. So it would be helpful if authors provide the biogeochemical data (at least in supp info) for each Niskin that is used for different light incubations.

3. How does the "average" intensity of light estimated. Were the light measurements continuous or monitored n times during the day?

4. While changing the light conditions, some density filters were used. Was the wavelength, which these filters block, was also estimated? Do they block the same fraction wavelength for all wavelength?

5. POC:PON ratio could be close to the Redfield ratio but the Carbon upatake:N2 fixation ratios (Fig. 4b) are surprising. CF:NF ratios can be upto three order magnitude higher even in tricho bloom conditions, where highest N2 fixation rates were measured (Gandhi et al., 2011). In not very active N fixation regions, this ratio could be even higher. This is simply because most of photoautotroph fix C but not all can N2. So C fixation to N (NO3+NH4+Urea+N2 fixation) uptake ratios would be close to Redfield. I would suggest the authors to look for the hypotheses presented on page 11 (lines 18-23) and explanations at several other places.

6. Provide an estimate of fraction of released DON and released inorganic N (ammonium) uptake by non-diazotrophs.

Minor Comments:

Page 1: Title should be revised as "field Trichodesmium" reads a bit awkward. I would suggest: "The effect of light on N2 fixation and net nitrogen release in a field study"

Page 3, line 8: Light is an ultimate source of energy for everything not only to photoautotrophs. Revised this sentence.

Page 4, line 9: Not most but only some NF rates have used 15N techniques, most have used Acetylene reduction assay, see Table 3 in (Singh et al., 2013), Table 5 in (Capone et al., 2005) and Table 4 in (Benavides and Voss, 2015).

Line 10: "The 15N. . . . . . .into account". 15N enrichment is taken into account as can be seen the equation (6) in (Montoya et al., 1996): the AN2 takes care of the enrichment. I think authors mean the released 15N-TDN during the incubation is not taken into the account and hence the underestimation.

Line 16-17: Contribution of N2 fixation to export production can be upto 92% during Trichodesmium bloom (Gandhi et al., 2011; Kumar et al., 2017)

Line 21: "reports" should be replaced by "has reported"

Page 5: Line 5: Could 4000 cells/L be called bloom?

Line 16: Were the nutrient samples filtered through 0.2 um filter? Were these measured at both the start and end of the incubations?

Line 22 and elsewhere: Reference format should be same throughout

Page 6, line 1: were should be replaced by was

Line 2: delete a

Line 5: put space after 40

Line 11: micron symbol throughout should be used rather than u

Line 22: (Mohr et al., 2010) is the original reference

Line 22: Were incubation done single, duplicates or triplicates?

Page 7, line 4: Were 13C and 15N2 added in the same bottles?

Line 7: Perhaps 0.7 $\mu$m pore size should be mentioned.

Line 8: It would be a surprise if the authors were able to filter 4.5 L water on single 25 mm GFF filter? Line 19: There is no Ik in equation (1)

Page 7, line 12: replace classical by typical

Line 14: How the average value of PAR calculated?

Line 18: 34.6 salinity is not really high. It is normal in open oceans

Line 23: "thus . . . . . . . . . . ...in all the experiments" can be deleted as preceding part of the sentence implicitly states the same.

Page 9, line 2: how many samples were taken to obtain the standard deviation and mean Line 3: "two order of magnitude" is not quite true.

Line 4: "Detail" should be replaced by "detailed"

Line 6: biomass should be replaced by abundance or cells

Line 18: If at t=0, POC was same in all the light experiments, then how does POC decrease with light within 24 hrs so rapidly. With this logic, POC concentration will be drastically different during the evening and in the morning in the ocean.

Page 10, line 8: replace "was decreasing" by "decreased"

Line 12: Define this mentioned fraction. Is it the ratio of 15N TDN uptake by non-diazotrophs and total production of 15N TDN by diazotrophs. Or is it the ratio of 15N TDN uptake by non-diazotrophs and total (15N + 14N) uptake by non-diazotrophs.

Line 19: "locates" does not read properly. Revise the sentence.

Line 21: Replace "strong of" by "strong"

Line 24: (Gandhi et al., 2011; Kumar et al., 2017) could also be proper citations here.

Page 11, line 2: 15-40 m is confusing here. Does it mean Trichos are more abundance in 15-40 m compared to that in 1-15 m?

Line 14: this section (including the hypothesis presented) should be revised as suggested in the major comments.

Page 1, line 6: "sever" should be replaced by "severe"

Lines 9-13: Not clear what the authors want to convey in this sentence

Page 13, line 16: (Montoya et al., 1996) is the original reference

Line 16: Why do the used a different technique may lead to higher DDN release?

Page 14, line 4: Replace "recently" by "recent"

Table 1: Also provide P* values (as expressed by (Deutsch et al., 2007)) in a column

Table 2: Also provide the fraction of diazotrophic biomass to the total phytoplankton biomass

Fig. 1: Why is there so much fluctuations (variation) within minutes in PAR values? Were the conditions cloudy during incubations?

Fig. 4: Either use CF/NF or PP/NF. Be consistent.

Supp Table 1: Normally enrichment in 13C is much more than 15N. How much was 13C added, and how much would the approximate theoretical 13C enrichment at t=0?

References:

Benavides, M., Voss, M., 2015. Five decades of N2 fixation research in the North Atlantic Ocean. Front. Mar. Sci. 2, 1–20. doi:10.3389/fmars.2015.00040

Capone, D.G., Burns, J.A., Montoya, J.P., Subramaniam, A., Mahaffey, C., Gunderson, T., Michaels, A.F., Carpenter, E.J., 2005. Nitrogen fixation by Trichodesmium spp.: An important source of new nitrogen to the tropical and subtropical North Atlantic Ocean. Glob. Biogeochem. Cycles 19, 1–17. doi:10.1029/2004GB002331

Deutsch, C., Sarmiento, J.L., Sigman, D.M., Gruber, N., Dunne, J.P., 2007. Spatial coupling of nitrogen inputs and losses in the ocean. Nature 445, 163–167.

Gandhi, N., Singh, A., Prakash, S., Ramesh, R., Raman, M., Sheshshayee, M., Shetye, S., 2011. First direct measurements of N2 fixation during a Trichodesmium bloom in the eastern Arabian Sea. Glob. Biogeochem. Cycles 25.

Kumar, P., Singh, A., Ramesh, R., Nallathambi, T., 2017. N2 Fixation in the Eastern Arabian Sea: Probable Role of Heterotrophic Diazotrophs. Front. Mar. Sci. 4, 80.

Mohr, W., Grosskopf, T., Wallace, D.W., LaRoche, J., 2010. Methodological underestimation of oceanic nitrogen fixation rates. PLOS One 5, e12583.

Montoya, J.P., Voss, M., KÓŞhler, P., Capone, D.G., 1996. A Simple, High-Precision, High-Sensitivity Tracer Assay for N2 Fixation. Appl. Environ. Microbiol. 62, 986–993.

Singh, A., Lomas, M., Bates, N., 2013. Revisiting N2 fixation in the North Atlantic Ocean: Significance of deviations from the Redfield Ratio, atmospheric deposition and climate variability. Deep Sea Res. Part II Top. Stud. Oceanogr. 93, 148–158.

Please also note the supplement to this comment:
https://www.biogeosciences-discuss.net/bg-2017-198/bg-2017-198-RC1-supplement.pdf

————————————————

---

## Author Comment (AC1) · 4 Sep 2017

The comment was uploaded in the form of a supplement:
https://www.biogeosciences-discuss.net/bg-2017-198/bg-2017-198-AC1-supplement.zip

———————————————————

---

## Author Comment (AC2) · 4 Sep 2017

The comment was uploaded in the form of a supplement:
https://www.biogeosciences-discuss.net/bg-2017-198/bg-2017-198-AC2-supplement.zip

---

## Referee Comment (RC2) · Anonymous Referee #3 · 5 Sep 2017

This manuscript looks at irradiance vs nitrogen fixation with an emphasis on Trichodesmium. This is a highly relevant topic and the authors approached the topic with a set of measurements that promised to increase our knowledge. Unfortunately at this moment, there is some missing information which makes it hard to evaluate the findings of this study. These are listed under major comments with additional items for consideration listed under minor comments

Major comments This manuscript does not include statistical analysis of the data which makes it difficult (impossible) to draw conclusions about the some of the measurements, such as Figure 5 which shows diazotroph derived nitrogen release rates.

It is not clear how the authors are defining a bloom of Trichodesmium bloom. I am aware of Trichodesmium accumulations in the form of slicks which are visually ob-

served, but a bloom to me is prolonged and active growth which should be validated

Elemental analysis of POC and PON is missing from the methods section. I suspect it derives from the15N-PON analysis but this should be discussed

The uncertainty associated with the light levels should be provided, particularly since the irradiance experiments are a critical component of the manuscript The authors mention 92, 54, 28, 14, 8, 1% but there will be variability associated with all of these values and the authors should say whether it is plus/minus 5%, 10% etc.

There is no mention of monitoring the temperature inside each of the incubators. If the incubators were plumbed with surface seawater then this can easily heat by >1oC and this will have an affect on the rates of carbon and nitrogen fixation.

Its not clear to me why all of the rates are attributed to Trichodesmium when the experiments were conducted on natural assemblages of mixed diazotrophs.

Why does Figure 1 show PAR of 4000 uE? I was under the impression that maximum sunlight was approx. 2500 uE.

Minor comments Page 2 Line 11 "NF pathway was likely preferentially blocked under low light to conserve energy for photosynthesis, thus, there is a metabolism tradeoff between carbon and nitrogen fixation pathways under light stress." I disagree with the wording of this statement. I think it is more likely that there is insuffiicent energy from photosynthesis at low light levels to support nitrogen fixation

Line 13 Define short-term light change. Is short-term <1 h or less than 1 day

Page 3 Bell and Fu (2005) observed an increasing NF rates. remove 'an'

Page 7, Line 1. Its not clear to me how you measured 15N-TDN.

Page 8 Section 2.7 How much confidence do you have in this filtration method to evaluate the transfer of DDN to no-diazos

Page 8 Line 23 What confidence do you have that it is T. thiebautii

Page 8, Line 24. I am not sure I follow the link between bulk POC and Trichodesmium-POC

Page 9 Line 1-6 I suggest moving the water-column nitrogen fixation rates to the previous section on environmental conditions

Page 9 Line 11-14 This should be in the same section as the NF-I

Page 9 How long were the incubations? The changes in POC are substantial and you should compare the increase in POC with the 13C-derived rate of productivity to make sure they agree.

Page 10 Section 3.4 This section cannot be included without statistical analysis

Page 10 I am not sure I follow your argument that the high light demand by Trichodesmium to fuel nitrogen fixation also help mitigate the problems caused by creating oxygen.

Page 11 Line 9 Did you ever consider conducting your incubations in situ? This would provide the light gradient you are after and as long as you are within the mixed layer then temperature would be constant (hopefully). I realize you lowest light levels might not attainable, but you should be able to cover 25-100% light levels.

Page 12 Line 14-24 I am not sure of the relevance of this paragraph to this study

Table 1 I increasingly see NOx being reported in the ocean literature and I dislike it application for describing nutrients due to the ambiguity. Report what was measured i.e. nitrate, nitrite. . .

Figure 2 Given the presence of other diazotrophs, how do you attribute the measured rates to Trichodesmium

Figure 3. I suspect the x-axis shows PAR equivalent to 92, 54, 28, 14, 8, 1% of the daily

averaged value, but this does not highlight the much higher intensities experienced. In Figure 1 you show PAR attaining values of 4000 uE and if this is true, it needs to be reflected.

---

## Author Response (AR1)

**Reply to SC-1**

From the perspective of a modeller with an interest in *Trichodesmium*, this is a useful paper. One reference you may have missed is Oliver et al. (2012). It would be interesting to see a comparison of the results in your manuscript with the results you'd expect given the energetic cost of nitrogen fixation versus nitrate uptake, and the energetic cost of chlorophyll production. Oliver et al. (2012) would be a useful resource to help make that comparison.

**Author response:**

Thanks for your appreciation and it is a good suggestion to do such comparison. In this study, we did not perform nitrate assimilation and chlorophyll dynamic monitor measurements, so it is hard to fulfill this target. However, in this informative paper, we got many helpful cellular metabolism physiology behaviors on energy allocation between different processes and added that information in our discussion part (Page 12 line 21-25 and Page 13 line 1-2). Regarding to the energetic cost of nitrogen fixation versus nitrate uptake, Eichner et al. (2014) had given a detailed study. Hope our founding could help the modellers.

Eichner, M., Kranz, S. A., and Rost, B.: Combined effects of different $CO_2$ levels and N sources on the diazotrophic cyanobacterium *Trichodesmium*, Physiologia plantarum, 152, 316-330, 2014.

**Reply to RC-1**

**Besides DDN release, what new in our paper are 1) this is the first data report of $N_2$-fixation irradiance curve with precise $I_k$ for *Trichodesmium*, particularly, in the field, 2) simultaneous measurements of C/$N_2$ fixation reveals light effect on *Trichodesmium*'s metabolism, 3) we applied the most advanced $^{15}N_2$ pre-enriched seawater method for $N_2$-fixation and DDN release**

**Referee #1, major comment #1**

Diazotroph derived nitrogen (DDN) release increases with increase in light intensity at S0320 (Fig. 5 a) but there was no variation at the other two stations (Fig. 5b,c). However, % of the total NF release always decreases. On the other hand, $N_2$ fixation increases with increase in light intensity and saturates at some

point (Fig. 4a). Put all these pieces of information together, it appears that it becomes difficult to say what role light play in DDN release. Diazotrophs would release N anyways, so what is the role light (they would release even if put them in absolute dark). Therefore, the discussion provided in the section 4.3 is not convincing

**Author response:**

Reviewer is correct about the light does not regulate the absolute amount of release and diazotrophs would release N anyway. However, the % release is an indication of budget or balance of N in cell. To discuss the physiological status for DDN redistribution, % release is a proper indicator. Stand on this point, light regulate the percent retention of fixed N in *Trichodesmium*.

According to this comment, we added two sentences to address the properness of using % release instead of absolute amount of release to represent the physiology status of *Trichodesmium* (Page13 line 9-13).

**Referee #1, major comment #2**

Were all the incubations samples at each station taken from the same Niskin Bottle? I believe not because of bottle capacity. As we know the sample (especially when the cell abundance is high) in different Niskin bottles could not be homogeneous although taken from the same depth and the same CTD. If the cell abundances were different in different light incubations to start with, then the rates would be different because of cells and not because of light. So it would be helpful if authors provide the biogeochemical data (at least in supp info) for each Niskin that is used for different light incubations

**Author response:**

All the incubations samples were taken from the same cast (same depth) but not same bottle. Six samples for *Trichodesmium* abundance may come from different Niskin bottles, however, results in Table 1 showed that the *Trichodesmium* abundance varied within 25% revealing consistency. On the other hand, NF rates varied from 400 to 2000% under different light conditions (n=3 for each light level). Although samples came from different Niskin bottles, such highly variable NF rates over irradiance should reflect mainly the physiological response of *Trichodesmium* to light. The heterogeneity was included in the error bar of NF measurements.

**Referee #1, major comment #3**

How does the "average" intensity of light estimated. Were the light measurements continuous or

monitored n times during the day?

**Author response:**

The light measurements were continuously monitored at ten second interval over the entire cruise. The "average" intensity of light was the averaged value of measured PAR larger than $1\mu E\ m^{-2}\ s^{-1}$ during the incubation period. In this version we added more information regarding the light monitoring and estimation of "average" light intensity.

**Referee #1, major comment #4**

While changing the light conditions, some density filters were used. Was the wavelength, which these filters block, was also estimated? Do they block the same fraction wavelength for all wavelength?

**Author response:**

This question was raised due to our unclear description. We followed published papers (Fernandez et al., 2013; Rijkenberg et al., 2011; Mourino-Carballido et al., 2011) to simulate lights by using Lee neutral density and blue (061 Mist blue; 172 Lagoon blue) filters. The neutral density filter blocked the same fraction wavelength while the blue filters prefer to block long wavelength and transmit more blue light. We added more descriptions in this version for light manipulation Material and Methods.

**Referee #1, major comment #5**

POC:PON ratio could be close to the Redfield ratio but the Carbon upatake:$N_2$ fixation ratios (Fig. 4b) are surprising. CF:NF ratios can be up to three order magnitude higher even in tricho bloom conditions, where highest $N_2$ fixation rates were measured (Gandhi et al., 2011). In not very active N fixation regions, this ratio could be even higher. This is simply because most of photoautotroph fix C but not all can $N_2$. So C fixation to N ($NO_3^+NH_4^+$Urea $N_2$ fixation) uptake ratios would be close to Redfield. I would suggest the authors to look for the hypotheses presented on page 11 (lines 18-23) and explanations at several other places.

**Author response:**

As mentioned by previous studies (Mulholland, 2007), *Trichodesmium* are known to exhibit a higher C:$N_2$ fixation ratios. Possible reasons were presented in the "Discussions 4.2" including the non-diazotrophs carbon fixation and other bioavailable N uptake mentioned by the reviewer. Indeed, in Gandhi et al. paper, most of observed CF:NF ratios were much higher than the Redfield ratio, however, in the surface bloom

condition (Station NF6), the NF rate in surface water was 1125 nM N h$^{-1}$ and CF rate was 4594 nM C h$^{-1}$, with a CF: NF ratio of ~4, even lower than the Redfield ratio.

In fact, in pure culture experiments of various diazotrophs including *Trichodesmium* (Berthelot et al., 2015), CF:NF ratios (1.8-5.6) were low and quite close to the POC:PON ratio (3.8-5.5) of cultured biomass. In our field study, *Trichodesmium* abundance was up to 4227 trichomes. L$^{-1}$, the measured CF:NF ratios (9.3) at *in situ* light also matched with the initial POC:PON ratio (6.4). Consistency among aforementioned studies suggested that CF/NF should not be particularly high.

According to this comment, we added more discussions (page 12 Lines 8-13).

**Referee #1, major comment #6**

Provide an estimate of fraction of released DON and released inorganic N (ammonium) uptake by non-diazotrophs.

**Author response:**

Actually, the Fig.6 was a simply flux estimation of released DDN (both DON and DIN) uptake by non-diazotrophs at surface condition. The basic procedures and principle were present in "Material and Methods 2.7". Unfortunately, the method for isotopic composition of low level NH$_4^+$ was not established in our laboratory. In this study, the fractions of released DDN (both DON and DIN) uptake by non-diazotrophs were only around 5% for at all three stations under *in situ* light. To separate DDN into DON and DIN fractions is an interesting idea indeed, but, impossible for our laboratory at current stage.

**Referee #1, minor comments:**

Page 1: Title should be revised as "field *Trichodesmium*" reads a bit awkward. I would suggest: "The effect of light on N$_2$ fixation and net nitrogen release in a field study".

**Author response:**

We would like to keep *Trichodesmium* in the title to highlight its importance. The new tile is "Light effect on N$_2$ fixation and net nitrogen release of *Trichodesmium* in the field".

Page 3, line 8: Light is an ultimate source of energy for everything not only to photoautotrophs. Revised this sentence.

**Author response:**

We changed to "light is the primary energy source".

Page 4, line 9: Not most but only some NF rates have used $^{15}$N techniques, most have used Acetylene reduction assay, see Table 3 in (Singh et al., 2013), Table 5 in (Capone et al., 2005) and Table 4 in (Benavides and Voss, 2015).

Line 10: "The $^{15}$N. . .. . ..into account". $^{15}$N enrichment is taken into account as can be seen the equation (6) in (Montoya et al., 1996): the $N_2$ takes care of the enrichment. I think authors mean the released $^{15}$N-TDN during the incubation is not taken into the account and hence the underestimation.

**Author response:**

Yes. The description is now "In most NF rates measurements that via incorporation of $^{15}N_2$ into

particulate organic N (PON), the $^{15}$N enrichment in the dissolved pool had not been taken into account, resulting in aforementioned potential underestimation of NF rates"

Line 16-17: Contribution of $N_2$ fixation to export production can be up to 92% during Trichodesmium bloom (Gandhi et al., 2011; Kumar et al., 2017)

**Author response:**

We added reference to highlight the importance of NF in export production (Page 4 Lines 18) in Introduction

Line 21: "reports" should be replaced by "has reported"

**Author response:**

Corrected

Page 5: Line 5: Could 4000 cells/L be called bloom?

**Author response:**

In station S0320, the abundance of *Trichodesmium* was up to 4227 trichomes/filaments. L$^{-1}$. Comparing with the previous study by Bonnet et al. (2016), it was under bloom condition.

Line 16: Were the nutrient samples filtered through 0.2 μm filter? Were these measured at both the start and end of the incubations?

**Author response:**

The nutrient samples were filtered through 0.45 μm cellulose acetate fiber and only measured at the initial
5   condition.

Line 22 and elsewhere: Reference format should be same throuhout
Page 6, line 1: were should be replaced by was
Line 2: delete a
10   Line 5: put space after 40
Line 11: micron symbol throughout should be used rather than u

**Author response:**

Corrected

15   Line 22: (Mohr et al., 2010) is the original reference

**Author response:**

Yes. We gave the credit to Mohr et al. (2010) in proper places. As for the $^{15}N_2$-enriched seawater preparation in this study we adapted the same device and procedure described in Shiozaki et al. (2015).

20   Line 22: Were incubation done single, duplicates or triplicates?

**Author response:**

All the incubations were triplicated. We added "triplicates" in proper places.

Page 7, line 4: Were $^{13}C$ and $^{15}N_2$ added in the same bottles?

25   **Author response:**

Yes. The $^{13}C$ and $^{15}N_2$ tracers were added in the same bottles.

Line 7: Perhaps 0.7 μm pore size should be mentioned.

**Author response:**

We added 0.7 μm into parenthesis following GF/F.

Line 8: It would be a surprise if the authors were able to filter 4.5 L water on single 25 mm GFF filter?
Line 19: There is no $I_k$ in equation (1)

**Author response:**

Reviewer is an expert indeed. At station D5 and A3, 4.5L PC bottles were used for incubations. While at S0320 with *Trichodesmium* bloom, 1.2L PC bottles were applied. We added more information in Material and Methods. We eliminated the description regarding $I_k$ in old sentence. A new sentence "The light saturation coefficient $I_k$ was defined as $N_m/\alpha$." was added showing the derivation of $I_k$.

Page 7, line 12: replace classical by typical

**Author response:**

Changed.

Line 14: How the average value of PAR calculated?

**Author response:**

See reply in the major comment #2.

Line 18: 34.6 salinity is not really high. It is normal in open oceans

**Author response:**

We added "relatively".

Line 23: "thus . . .. . .. . . in all the experiments" can be deleted as preceding part of the sentence implicitly

states the same.

**Author response:**

Deleted.

Page 9, line 2: how many samples were taken to obtain the standard deviation and mean Line 3: "two order of magnitude" is not quite true.

**Author response:**

Triplicated samples results were used. The description is now "The NF rate at the blooming station was 30-40 times higher than that of the two non-bloom stations.".

Line 4: "Detail" should be replaced by "detailed"

**Author response:**

Corrected

Line 6: biomass should be replaced by abundance or cells

**Author response:**

We changed "biomass" to "trichome".

Line 18: If at t=0, POC was same in all the light experiments, then how does POC decrease with light within 24 hrs so rapidly. With this logic, POC concentration will be drastically different during the evening and in the morning in the ocean.

**Author response:**

The POC concentration ranged from $10.9 \pm 1.0$ μM to $17.3 \pm 1.2$ μM, which is 100 times higher than our EA-IRMS detection limit (1 μgC). The discrepancies among various light incubations were 0.5 -3.9μM, which is apparently measurable. From our triplicates, the discrepancy is significant statistically. Moreover, such change happened in both POC and PON. The detectable POC/PON change may be due to the high biomass in blooming condition. Nevertheless, changes in *Chl*-a within 24 hrs incubation were reported in many previous studies.

Page 10, line 8: replace "was decreasing" by "decreased"

**Author response:**

Replaced.

5    Line 12: Define this mentioned fraction. Is it the ratio of $^{15}$N TDN uptake by nondiazotrophs and total production of $^{15}$N TDN by diazotrophs. Or is it the ratio of $^{15}$N TDN uptake by non-diazotrophs and total ($^{15}$N + $^{14}$N) uptake by non-diazotrophs.

**Author response:**

This mentioned fraction is the ratio of $^{15}$N TDN uptake by non-diazotrophs and total DDN flux. The
10   description is now "After 24 h incubation, the DDN transfer rates (transferred to the non-diazotrophic plankton) were 18.6± 3.6, 0.5 ± 0.3 and 0.7 ± 0.5 nM N d$^{-1}$ corresponding to 5% ± 1%, 4% ± 3% and 5% ± 4% of total NF (net plus dissolved), respectively, for Stas. S0320, D5 and A3 (Fig. 6)."

Line 19: "locates" does not read properly. Revise the sentence.

15  **Author response:**

Changed to "fell within".

Line 21: Replace "strong of" by "strong"

**Author response:**

Corrected.

Line 24: (Gandhi et al., 2011; Kumar et al., 2017) could also be proper citations here.

**Author response:**

Added.

25  Page 11, line 2: 15-40 m is confusing here. Does it mean Trichos are more abundance in 15-40 m compared to that in 1-15 m?

**Author response:**

The sentence was changed to "By taking into account light extinction coefficient of seawater, the maximum depth for *Trichodesmium* to perform NF would be shallower than 15-40m."

Line 14: this section (including the hypothesis presented) should be revised as suggested in the major comments.

**Author response:**
We revised as suggested following the major comment #2.

Page 12, line 6: "sever" should be replaced by "severe"

**Author response:**

Corrected.

Lines 9-13: Not clear what the authors want to convey in this sentence

**Author response:**

The sentence was changed to "Since our experiments of short-term light manipulation in our experiments resembles the natural variation of irradiance, such metabolism tradeoff between carbon and nitrogen fixation under low light for *Trichodesmium* may happen frequently and widespread in the field, such as cloudy day and rainy day."

Page 13, line 16: (Montoya et al., 1996) is the original reference

**Author response:**

Added.

Line 16: Why do the used a different technique may lead to higher DDN release?

**Author response:**

This issue is thoroughly discussed in Berthelot et al. (2015). The main reason is in previous method DDN release were estimated by the comparison of the gross and net NF rate, the large uncertainty of gross NF

measured by acetylene reduction assays (ARA) and underestimation of net NF by the $^{15}N_2$ bubble method may overestimate the DDN release. While, the method used in this study was directly measure the recently fixed $^{15}N$ signal in dissolved pool.

Page 14, line 4: Replace "recently" by "recent"

**Author response:**

Corrected.

Table 1: Also provide P* values (as expressed by (Deutsch et al., 2007)) in a column

**Author response:**

In this study, three sampling stations are all oligotrophic surface ocean the bioavailable nitrogen concentrations only around 10 nM, and the SRP concentrations were also at nM level, except in bloom station S0320. We has listed all measurable data in table for readers to estimate P* values. We hope reviewer can accept our answer.

Table 2: Also provide the fraction of diazotrophic biomass to the total phytoplankton biomass

**Author response:**

This is a good idea; unfortunately, there is no feasible way to separate diazotrophic biomass from bulk biomass now.

Fig. 1: Why is there so much fluctuations (variation) within minutes in PAR values? Were the conditions cloudy during incubations?

**Author response:**

Yes, it was caused by floating cloud and cloud cover.

Fig. 4: Either use CF/NF or PP/NF. Be consistent.

**Author response:**

We used CF/NF. It is consistent now.

Supp Table 1: Normally enrichment in $^{13}C$ is much more than $^{15}N$. How much was $^{13}C$ added, and how much would the approximate theoretical $^{13}C$ enrichment at t=0?

**Author response:**

In this study, $^{13}C$-labeled sodium bicarbonate (99 atom% $^{13}C$; Cambridge Isotope Laboratories) was added
5   to each bottle at a final tracer concentration of 70 μmol L$^{-1}$. The enrichment of $^{13}C$ finally up to 3.5% at t=0. We added this information in this revision.

**Reply to RC-3**

**Referee #3, major comment #1**

This manuscript does not include statistical analysis of the data which makes it difficult (impossible) to draw conclusions about the some of the measurements, such as Figure 5 which shows diazotroph derived nitrogen release rates.

**Author response:**

In this version, we did statistical analyses and presented results in proper places to support our statements.

For example, the R squares of fitted NF-I curves were 0.92, 0.71 and 0.95 at station S0320, A3 and D5, respectively, in Fig.3. For Fig.4a, the R squares of fitted CF-I curve was 0.90 at station S0320. For Fig.4b, the linear regression of CF/NF versus PAR ($<410$ $\mu E$ $m^{-2}$ $s^{-1}$) showed that the slope was -0.023, R squares value was 0.72 and the P was 0.0005.

**Referee #3, major comment #2**

It is not clear how the authors are defining a bloom of *Trichodesmium* bloom. I am aware of *Trichodesmium* accumulations in the form of slicks which are visually observed, but a bloom to me is prolonged and active growth which should be validated

**Author response:**

Yes, we saw the form of slicks from naked eyes at station S0320 (0° N,142° W ), where the *Trichodesmium* abundance was up to 4227 trichomes/filaments. $L^{-1}$. The surface CF rate was up to 3.6 $\mu M \ L^{-1} \ d^{-1}$ and the surface NF rate was 391 nM $L^{-1} \ d^{-1}$, resulting in a turnover time (C, N based) of ~4-5 days.

Comparing with the previous study by Bonnet et al. (2016), it was under bloom condition. Actually, the bloom area was not limited at S0320, the bloom covered >1x1 degree (0° N,141° W ). Unfortunately, we do not have proper remote sensing algorithm to identify the size of bloom specifically for *Trichodesmium*.

**Referee #1, major comment #3**

Elemental analysis of POC and PON is missing from the methods section. I suspect it derives from the $^{15}$N-PON analysis but this should be discussed

**Author response:**

We added the detailed measurement method of POC and PON in this version manuscript (Page 7 line 10 - 12).

**Referee #3, major comment #4**

The uncertainty associated with the light levels should be provided, particularly since the irradiance experiments are a critical component of the manuscript The authors mention 92, 54, 28, 14, 8, 1% but there will be variability associated with all of these values and the authors should say whether it is plus/minus 5%, 10% etc.

**Author response:**

The variability of daily irradiance is 61% - 83% according to on-deck PAR record of a minute interval. We provided this uncertainty value in this version.

**Referee #3, major comment #5**

There is no mention of monitoring the temperature inside each of the incubators. If the incubators were plumbed with surface seawater then this can easily heat by >1°C and this will have an effect on the rates of carbon and nitrogen fixation.

**Author response:**

All bottles were incubated in on-deck incubator with rapid pumping surface water flow-through (~60 L min$^{-1}$). The total volume of 10 incubator tanks was ~540 L, so the water turnover time of every individual incubator was ~9 min.

In fact, the water temperature in incubation tank is slightly higher (<1 degree) than in situ surface sea water. Additionally, the forcing from temperature on the variability pattern can be ignored since all bottles were in the same temperature situation.

**Referee #3, major comment #6**

Its not clear to me why all of the rates are attributed to *Trichodesmium* when the experiments were conducted on natural assemblages of mixed diazotrophs.

**Author response:**

We did not attribute to *Trichodesmium* except for the Sta. S0320 with *nif*H gene of *Trichodesmium* >98.8%. For A3 and D5, we assumed the pattern of light effect is mainly driven by *Trichodesmium* according to their *nif*H gene abundance of >89% and >96%.

**Referee #3, major comment #7:**

Why does Figure 1 show PAR of 4000 µE? I was under the impression that maximum sunlight was approx. 2500 µE.

**Author response:**

We are really very grateful for review's this question. In this study, we have both spherical 4π photosynthetically available radiation (PAR) sensor (QSL-2100; Biospherical instruments Inc.) and flat 2π PAR sensor (PQS 1 PAR Quantum Sensor, Kipp & Zonen). According to this comment, we applied irradiance data from PQS for all plots. Results are more consistent with available reports.

**Referee #1, minor comments:**

Line 11 "NF pathway was likely preferentially blocked under low light to conserve energy for photosynthesis, thus, there is a metabolism tradeoff between carbon and nitrogen fixation pathways under light stress." I disagree with the wording of this statement. I think it is more likely that there is insufficient energy from photosynthesis at low light levels to support nitrogen fixation.

**Author response:**

In this version, we changed the wording and added one paragraph (Page 12, line 23-25 and Page 13, line 1-4) to elucidate the physiological mechanism of energy reallocation under light stress condition. "The proper allocation and utilization of energy (ATP) and reductant (NADPH) among various cellular processes determines the growth rate of *Trichodesmium*. Light-dependent reactions of photosynthesis are the major pathway to produce these molecules. In cyanobacteria, both respiratory and photosynthetic electron transport occur in the thylakoid membrane and compete for the electron transport chain (Oliver et al., 2012). When light intensity decreases, the light-dependent reactions of photosynthetic activity would decrease concurrently, resulting in reduced production of ATP and NADPH and increased activity of respiration. The negative feedback of POC consumption leads to more ATP and NADPH being reallocated to CF process, and in turn, the NF process would be down-regulated.".
We now rephrased our statement to "We hypothesized that under low light stress, *Trichodesmium* physiologically prefer to allocate more energy for CF to alleviate the intensive carbon consumption by respiration.".

Line 13 Define short-term light change. Is short-term <1 h or less than 1 day

**Author response:**

We added a parenthesis with less than 24h.

Page 3 Bell and Fu (2005) observed an increasing NF rates. remove 'an'

**Author response:**

Corrected.

Page 7, Line 1. It's not clear to me how you measured $^{15}$N-TDN.

**Author response:**

The detailed information about $^{15}$N-TDN measurement is now in Material and Methods 2.6.

Page 8 Section 2.7 How much confidence do you have in this filtration method to evaluate the transfer of DDN to no-diazotrophs

**Author response:**

We have confidence about the transfer of DDN to non-diazotrophs. First, the colony counting shows no heterogeneity among water samples (n=6) from the same depth. Secondly, the discrepancy of $\delta^{15}$N value between two treatments ranged from 10 to as high as 770‰. This is much larger than the standard deviation of triplicates.

Page 8 Line 23 What confidence do you have that it is *T. thiebautii.*

**Author response:**

We used Nikon Eclipse 50i optical microscope to count the abundance and constrain the species of

[Figure]

*Trichodesmium.* We have confidence about the *T. thiebautii.*

[Figure]

Page 9 Line 1-6 I suggest moving the water-column nitrogen fixation rates to the previous section on environmental conditions

**Author response:**

In this version, we adjust this section to 'Environmental conditions'

Page 9 Line 11-14 This should be in the same section as the NF-I

**Author response:**

In this version, we adjust this section to 'NF-I curves'

Page 9 How long were the incubations? The changes in POC are substantial and you should compare the increase in POC with the 13C-derived rate of productivity to make sure they agree.

**Author response:**

The incubations last for 24 hours. We added illustrations of comparison in table 2 in this version. The increases in POC were comparable with the $^{13}$C-derived rate of productivity at light saturation conditions (larger than 400 $\mu E\ m^{-2}\ s^{-1}$). The two values were quite matched since the respiration rate was low. While the light intensity was under saturation value, the discrepancy increased due to the increasing of respiration rate.

Page 10 Section 3.4 This section cannot be included without statistical analysis

**Author response:**

In this version, we added statistical analyses at proper places to support our statements.

Page 10 I am not sure I follow your argument that the high light demand by *Trichodesmium* to fuel nitrogen fixation also help mitigate the problems caused by creating oxygen.

**Author response:**

Since the oxygen evolved by photosynthesis is toxic to nitrogenase, when *Trichodesmium* conduct NF processes, much energy was allocated to consume the oxygen to create the anaerobic microenvironment. Our statement is "The high energy requirement of *Trichodesmium* is not only for breaking the strong of triple bond of the $N_2$ molecule, but also for numerous strategies, such as high respiration rates and the Mehler reaction, to protect the sensitive nitrogenase against the oxygen evolved by photosynthesis during

day time (Kana 1993)".

Page 11 Line 9 Did you ever consider conducting your incubations *in situ*? This would provide the light gradient you are after and as long as you are within the mixed layer then temperature would be constant (hopefully). I realize you lowest light levels might not attainable, but you should be able to cover 25-100% light levels.

**Author response:**

It's a good suggestion and actually many studies prefer *in situ* incubations now. But in this study, due to the cruise time schedule, we have no enough ship time to conduct *in situ* incubations. We hope to do it in future cruises.

Page 12 Line 14-24 I am not sure of the relevance of this paragraph to this study

**Author response:**

In this version, we added reasons to bridge the logic gap (Page 13, Line 11-15).

"Our results demonstrated that light does not directly regulate the absolute amount of DDN release. However, to discuss the physiological status for DDN distribution in dissolved pool and particulate pool (mainly *Trichodesmium*), the proportion of DDN released into the dissolved pool is a proper indicator. In this study, the increased proportion of DDN in the dissolved pool as the decrease of light intensity suggested that physiology status of diazotrophs modulated by light could take control on the DDN release process.".

Table 1 I increasingly see NOx being reported in the ocean literature and I dislike it application for describing nutrients due to the ambiguity. Report what was measured i.e. nitrate, nitrite. . .

**Author response:**

Changed as requested.

Figure 2 Given the presence of other diazotrophs, how do you attribute the measured rates to *Trichodesmium*

**Author response:**

We cannot exclude the NF by other diazotrophs, so we adapted the description of 'particulate NF' in proper places to be more precise

Figure 3. I suspect the x-axis shows PAR equivalent to 92, 54, 28, 14, 8, 1% of the daily averaged value, but this does not highlight the much higher intensities experienced. In Figure 1 you show PAR attaining values of 4000 uE and if this is true, it needs to be reflected.

**Author response:**

Reviewer is right. In this version, we use the $2\pi$ PAR sensor we added the standard deviation of the average of light recorded on-deck for the day of incubation (Page 15, Line 17).

**Reference**

[revised manuscript text omitted]

---

## Author Response (AR2)

**Reply to Referee#3**

**comment #1**

The authors acknowledge that the temperature in the incubators exceeded that of surface seawater, but that the temperature increase was <1°C. This indicates to me that temperature measurements were made, and therefore they should be reported. This might seem pedantic, but it becomes critically important as the authors need to demonstrate that the observed effects of light intensity are independent of temperature.

**Author response:**

We agree the referee's opinion that we need to demonstrate the observed effects of light intensity are independent of temperature. In this version we added 'The surface cooling seawater was connected with incubators in parallel to keep the temperature variations in six incubators were synchronous. Thus, temperature was not a variable parameter that could influence the variability of final rates.' in text (page 7, line 9-11)

**comment #2**

The authors also need to improve their statistical reporting as 'R squares' is incorrect.

**Author response:**

Corrected.

---

## Editor Decision (ED2)

**Title:** Effect of light on $N_2$ fixation and net nitrogen release by Trichodesmium in a field study

**Abstract.** Dinitrogen fixation (NF) by marine cyanobacteria is  an important pathway to replenish the oceanic bioavailable nitrogen inventory. Light is the key to modulate NF, however, field studies  investigating light response curve (NF-I curve) of NF rate and the effect of light on diazotroph derived nitrogen (DDN) net release are  relatively sparse in the literature hamper  prediction by the models.  A dissolution method was applied using uncontaminated $^{15}N_2$ gas to examine how the light change may influence the NF intensity and DDN net release in the oligotrophic ocean. Experiments were conducted at stations with diazotrophs dominated by filamentous cyanobacterium *Trichodesmium spp.* in the Western Pacific and the South China Sea. The  effect of light on carbon fixation (CF) was measured in parallel using the $^{13}C$  tracer method specifically for a station characterized by *Trichodesmium* bloom. Both NF-I and CF-I curves showed *Ik* (light saturation coefficient) range of 193 to 315 $\mu E\ m^{-2}\ s^{-1}$ with  light saturation at around 400 $\mu E\ m^{-2}\ s^{-1}$. The proportion of DDN net release ranged from ~6% to ~50%  suggesting an increasing trend as the light intensity decreased. At the *Trichodesmium* bloom station, we found CF/NF ratio was light-dependent and the ratio started to increase as light was lower than the carbon compensation point of 200 $\mu E\ m^{-2}\ s^{-1}$. NF pathway was most likely  blocked under low light to conserve energy for photosynthesis, thus, there is a metabolism tradeoff between carbon and nitrogen fixation pathways under light stress. Results showed that short-term (<24h) light change modulates the physiological state, which subsequently determined the C/N metabolism and DDN net release  by *Trichodesmium*.  Reallocation of energy associated with the variation  in light intensity would be helpful for  prediction of global biogeochemical cycle of N by models  involving *Trichodesmium* bloom.